# Biomechanical study on the effect of atherosclerosis on the vulnerability of thoracic aorta, and it's role in the development of traumatic aorta injury

**Dénes Pauka**[1], **Viktor Soma Poór**[1], **Péter Maróti**[2,3], **Roland Told**[2], **Dénes Tóth**[1], **Tamás Tornóczky**[4], **Tamás F. Molnár**[5,6], **Gábor Simon**[1] *

**1** Department of Forensic Medicine, Medical School, University of Pécs, Pécs, Hungary, **2** 3D Printing & Visualisation Centre, University of Pécs, Pécs, Hungary, **3** Medical Skills Education and Innovation Centre, Medical School, University of Pécs, Pécs, Hungary, **4** Department of Pathology, Medical School, University of Pécs, Pécs, Hungary, **5** Department of Surgery, Petz A University Teaching Hospital, Győr, Hungary, **6** Medical Skills Education and Innovation Centre, Operational Medicine Group, Medical School, University of Pécs, Pécs, Hungary

☯ These authors contributed equally to this work.

* gabor.simon@aok.pte.hu

**Data Availability Statement:** All relevant data are within the paper and its Supporting Information files.

## Abstract

Traumatic aorta injury (TAI) is the second most common traumatic cause of death preceded only by head injuries, being responsible for 5% to 30% of all mortalities in high-speed deceleration injuries. Multiple external factors might play a role such as impact speed, impact direction, occupant location, and presence or lack of restraining safety mechanism. Apart from these external factors, also human biological factors can influence its development. Based on the data of scientific literature, age clearly plays a role in suffering TAI, but the role of atherosclerosis–as a disease affecting the structure of the aorta–is unknown. Biomechanical properties of tissue samples of 104 aorta specimens removed during the autopsy from the posterior (Group 'A') and lateral wall (Group 'B') of descending aorta were analyzed. Specimens were examined by a Zwick/Roell Z5.0 biaxial tester. The Young's modulus ($E$ (MPa)) was calculated using a linear regression procedure where the base of the elongation was the parallel length of the sample, the achieved maximal force ($F_{max}$ (N)), the elongation at the time of $F_{max}$ ($L_{max}$ (mm)), the force at the beginning of rupture ($F_{break}$ (N)), the elongation at the time of $F_{break}$ ($L_{break}$ (mm)) were registered. Specimens were categorized based on macroscopic and microscopic appearance. In the posterior (A) samples the difference between $L_{break}$ (p<0.001) and $L_{max}$ (p<0.001) was significant between the macroscopic group. $L_{break}$ (p = 0.009) and $L_{max}$ (p = 0.003) showed similar pattern in the lateral (B) samples. Comparing the histological groups by the measured parameters ($F_{max}$, $L_{max}$, $F_{break}$, $L_{break}$) showed a significant difference in the means (p<0.001, p = 0.003, p<0.001 respectively). The study demonstrated that atherosclerosis decreases the resistance of the aorta. The rupture occurs at lower force ($F_{max}$ and $F_{break}$), and at shorter elongation ($L_{max}$ and $L_{break}$) in case of the presence of atherosclerosis. This effect is most substantial if calcification is present: the resistance of aorta affected by calcification is only two-thirds on average

**Funding:** Project no. TKP2021-NVA-06 has been implemented with the support provided from the National Research, Development and Innovation Fund of Hungary, financed under the TKP2021-NVA funding scheme. It provides support for P.M. The work is related to the Thematic Excellence Program 2021—National Excellence Sub-program; Biomedical Engineering Project ("2020-4.1.1-TKP2020" The funders had no role in study design, data collection and analysis, decision to publish, or preparation of the manuscript.

**Competing interests:** The authors have declared that no competing interests exist.

compared to aorta affected by the early phase of atherosclerosis. This phenomenon can be clearly explained by the weakening structure of the tunica intima.

## Introduction

### Risk factors of traumatic aortic rupture

Approximately 1.3 million people die annually because of road traffic accidents [1]. Traumatic aorta injury (TAI) is the second most common traumatic cause of death preceded only by head injuries, being responsible for 5% to 30% of all mortalities in high-speed deceleration injuries [2–4]. Apart from traffic accidents, falling from height is also a common cause of TAI [5], the incidence of TAI in fatal cases of falling from height is 6% - 15% [6, 7].

All possible factors potentially contributing to the development of TAI must be considered in a forensic analysis. The resultant determined in the actual trauma is reducible to two components: the suffering human subject and objective external factors. Multiple external factors might play a role such as impact speed, impact direction, occupant location, and restraining safety mechanism [4, 8–10]. Individual differences due to person-specific biomechanical characteristics of tissues such as age, diseases, BMI (body-mass index) are to be considered in the vulnerability/resilience of the aorta in question as well. These characteristics are reflected in the macrostructure (wall diameter and thickness) and microstructure (fiber composition), affecting the composition and strength of tissues [11, 12].

Age clearly plays a role in developing TAI, as victims above 59 years of age have a three times higher risk of suffering thoracic aortic injury, especially in low impact speed [8, 9]. This can be explained by the macro and microstructural changes associated with age [12]. One of the pathologies which is connected to the aortic injuries in a contributory role is cystic media necrosis (CMN) [13] characterized by the local disappearance of elastic fibers in the arterial media and reduction in smooth muscle cell numbers [14]. However, there is no reliable data available about its causative role in traumatic aortic injuries, and also the CNM prevalence is low in the general population [14, 15].

Traumatic rupture of the aorta most probably originates in the intima layer [16–18], which lowers further the possibility of role of CNM in developing TAI, and also draws attention to the importance of atherosclerosis as the most common disease affecting the intima. Above the age of 30 years everyone is affected by some degree of aortic atherosclerosis [19, 20], while research in the age group of 12–33 years found that everyone had at least intimal thickening [21]. The role of atherosclerosis in the development of aortic aneurysms and subsequent spontaneous rupture is well established [22], but there is no reliable data available about the possible role of atherosclerosis in traumatic aortic injuries (TAI). Strassman suggested that atherosclerosis is not important in determining the development or the site of the aortic injuries, but his study was limited only to the evaluation of 72 cases of traumatic aortic injuries [16].

### Biomechanics of aorta

Blood vessels are exposed to multiple mechanical forces that are exerted on the vessel wall (radial, circumferential and longitudinal forces) or on the endothelial surface (shear stress) [23]. Circumferential and longitudinal (axial) stresses are the dominating principal stress of the aorta under normal physiologic conditions. Aorta is anisotropic, as the mechanical characteristics in longitudinal and circumferential directions differ from each other [24]. Collagen fibers are more densely aligned along the circumferential than the longitudinal direction, and

the stress causing rupture by exceeding wall strength is much lower in the longitudinal than circumferential direction [25].

Guinea et al. have shown in their biomechanical study involving 29 human descending aortae, that the mechanical behavior of the aortic wall decreases with age, and the tensile strength decreases abruptly beyond the age of 30. They found that in age below 30, the circumferential tensile strength exceeds longitudinal tensile strength (2.4 MPa ± 0.2 MPa versus 1.3 MPa ± 0.2 MPa), but this difference is smaller over the age of 46 (0.9 MPa vs 0.7 MPa). The failure stretch of the aorta also decreases with age, but only slightly–older samples represented 86% values of younger samples [26]. Garcia-Herrera et al. found similar age-related differences in the case of ascending aorta [27]. Animal experiments show that tissue stiffness is determined by the density of covalent cross-links between collagen molecules, and the decrease in cross-links–which is seen in the case of aortic enlargement–will decrease the stiffness and increase the risk of rupture [28, 29]. Mouse experiments also show the importance of elasticity and elastic fibers: elastic fiber defects increase stiffness and weaken the aortic wall [30].

Kozun et al. demonstrated that atherosclerosis increases the susceptibility of the thoracic aortic wall to delamination but does not affect its anisotropy [31].

## Mechanism and biomechanics of blunt traumatic aortic rupture

According to Chang et al., the most common location of TAI is the superior part of the descending aorta (33%), followed by the ascending aorta (26%), the inferior half of the descending aorta (21%) and the 2 cm long part of the aorta after the orifice of the left subclavian artery (16%) [32]. Other studies agree that the aortic isthmus is the most commonly injured part of the aorta [13, 17].

Different mechanisms are assumed behind the development of blunt aortic injuries, and most injuries involve a combination of different mechanisms [33]. Possible mechanisms include the stretching effect from sudden deceleration and acceleration of the heart (at the point where the more mobile aortic arch continues in the descending aorta), a sudden rise of intravascular pressure, implosion of the sternum which presses the aorta to the vertebral column ("osseous pinch"), the hyperflexion of the artic arch due to extreme chest compression, deflection of the aortic arch by the upwards displacement of mediastinal organs in lower chest wall impact ("shoveling effect"), the sudden elevation of intraabdominal pressure in abdominal impact ("water-hammer effect") [10, 33–35]. In the case of lateral impacts, lateral movement of the heart due to lateral deceleration and shock waves produced by the internal deformation of the chest wall leads to shear injury at the isthmus [21]. The main injury forces can be stretching, pinching, shearing, and humping [20].

The actual role of these different mechanisms is strongly determined by the injury scenario: in the case of a frontal collision and unrestrained occupant, aortic rupture is associated with thoracic and abdominal compression due to deceleration of the body at the moment when the driver's body impacts into the steering wheel. In case of a frontal collision and restrained occupants (especially in the case of front-driving passenger), the caudo-rostral hyperextension is an important mechanism: in this case, the carotid vessels pull the aortic arch upwards and forward due to inertia from the forward moving head, while at the same time the intercostal arteries fix the thoracic part of the aorta and pull it downwards [36].

The experiment of Stemper et al. on porcine aorta specimens showed that the rupture always starts in the intimal layer [18], which matches with the observations during autopsies of human TAI [16, 17]. Lundevall measured 10 aortas from autopsies and demonstrated that the tensile strength of the isthmus is lower than the tensile strength of the ascending aorta. He demonstrated that the resistant power of the cross-section of the aortic wall is 7400 g/cm$^2$ at

aorta ascendens, and 5250 g/cm$^2$ at aorta descendens, water hammer-effect will cause rupture at 196 G—286 G deceleration [37]. All specimens of Lundevall showed various degrees of acceleration without calcification, but Lundevall did not examine the role of severity of atherosclerosis.

Despite the fact that atherosclerosis has an impact on the mechanical properties of the aorta, the literature review highlighted that the connection between TAI and atherosclerosis has not been described before. Although it is known, that the composition of atherosclerotic plaques determines their ability to withstand mechanical load [23], it is not known whether they have a similar effect on the vulnerability of the whole vessel wall during longitudinal mechanical load typically emerge in high energy trauma. The aim of our study was to investigate the possible correlation between the mechanical properties (blunt force vulnerability) of the thoracic aorta and the severity of atherosclerosis.

## Materials and methods

### Study design

A biomechanical study was designed to investigate the effect of atherosclerosis on blunt force vulnerability of the thoracic aorta. The goal of the study was not mainly to describe the exact mechanical properties of the aorta, rather to make a comparison of vulnerability of aortas with various severity of atherosclerosis against forces typically occurring during a blunt TAI. Human aorta specimens were removed from cadavers, and their resilience was measured with a tensile tester. Tensile testing applies a static load on the ends of the specimen while the displacement is measured. The load divided by the cross-sectional area gives the stress and the displacement divided by the length over which displacement was measured gives the strain. The Young's modulus is then given by the stress divided by the strain [38]. Uniaxial testing of aorta specimens in the longitudinal direction was chosen as a testing method because this emulates the dominant direction of load (stress) in case of blunt TAI. The force at the failure of tissue (rupture) was recorded. The specimens were analyzed and grouped by macroscopic and microscopic appearance, and the results of tensile testing (force) was compared and statistically analyzed between the various macro- and microscopic groups.

### Ethical approval

Aorta samples were obtained from cadavers undergoing pathological and forensic autopsies All personal data was anonymized. Organs and tissues from cadavers can be removed for scientific purpose if the deceased does not object against it before death, and the data can be utilized freely for scientific and educational purposes without informed consent according to the 40. § (3) of the Hungarian act of Forensic Experts (2016.XXIX) and 220. § (1) of the Hungarian Healthcare act of 1997. The study was approved by the Regional Research Ethic Committee, Pécs (8836 –PTE 2021).

### Samples

The biomechanical properties of tissue samples of 104 aorta specimens from 52 autopsy cases removed during the autopsy were examined by a tensile tester. Autopsy cases suffering a high-energy trauma (e.g., traffic accident or falling from heights) were excluded to avoid the possibility of non-visible traumatic aorta injury. Cases showing any signs of putrefaction were also excluded. The average age at death of victims was 70 years (SD: ±11.83, min-max: 34–88). Thirty-two were males (age: 34–84 years, 66 years on average) and 20 females (age: 53–88, 76 years on average).

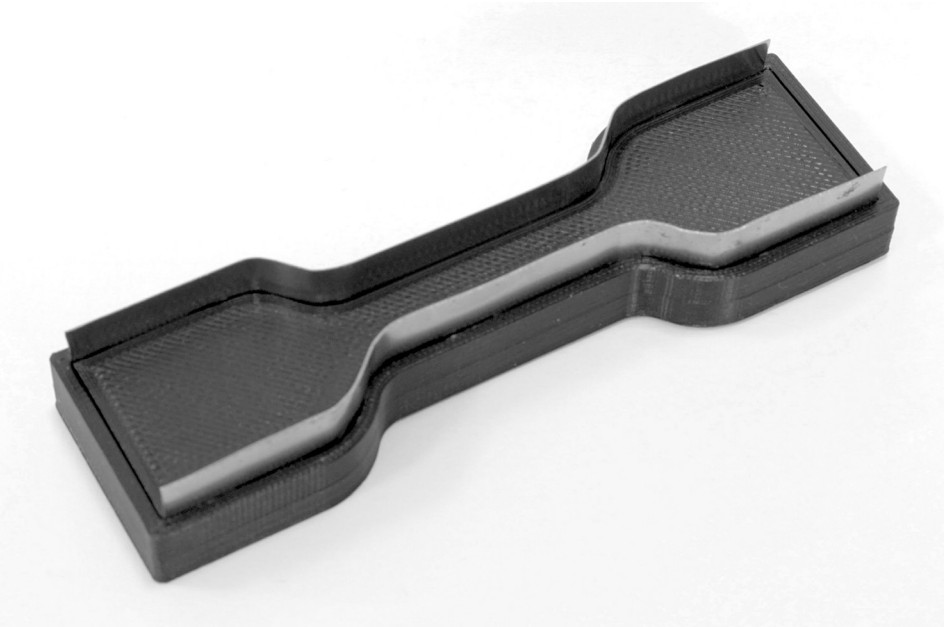

**Fig 1. Specimen removal tool.** To ensure consistent sampling a 3D cutting tool was prepared with steel cutting edges shaped as the ISO test sample.

Bodies were stored at 4˚C prior to the autopsy. Chest organs and aorta was removed in the conventional way during the autopsy, and after the standard longitudinal-posterior incision, two dog-bone-shaped samples were removed from each descending aorta at the same height starting from the level of the first intercostal artery take off with a specially designed tool (Fig 1). The total sample length was 10 cm, and the parallel length was 32.1 mm. One sample (Group 'A') was taken from the area between the orifices of the left and right intercostal arteries (posterior wall), and another sample (Group 'B') was cut from lateral to the left intercostal artery to the original incisional line (lateral wall) from all aortas (Fig 2). Time of death, time of obtaining the specimen, and time of measurement were recorded with an accuracy of minutes.

Ninety-nine samples were successfully measured and analyzed by their macroscopic appearance (one sample was damaged during sample removal, and four were discarded for processing technical failures). Ninety-eight samples were analyzed by their microscopical appearance (one was excluded because of damage to the sample during histological slide preparation).

## Measurement of sample thickness

Sample thickness was measured by a MecMesin MultiTest dV (capacity: 0–2.5 kN; Mecmesin, Slinford, UK) motorized force tester combined with a Mecmesin advanced force gauge (AFG) AFG-50 (measurement range: 0 N—50 N, resolution 0.01 N, accuracy: ±0.005 N; Mecmesin, Slinford, UK). A Mecmesin radius test rod with 8 mm diameter (Part. No. 432–354, Mecmesin, Slinford, UK) was applied to the AFG, and the displacement of the force tester was zeroed as the bottom of the test rod reached the bottom plate of the force tester. The test rod was elevated by manual control, then the aorta sample was laid down to the bottom plate of the force tester by its exterior surface (tunica adventitia). The test rod was moved downwards against the part of the aorta which microscopically showed no atherosclerosis (or the thinnest part of the whole sample was covered by an atherosclerotic plaque). Thickness was determined by

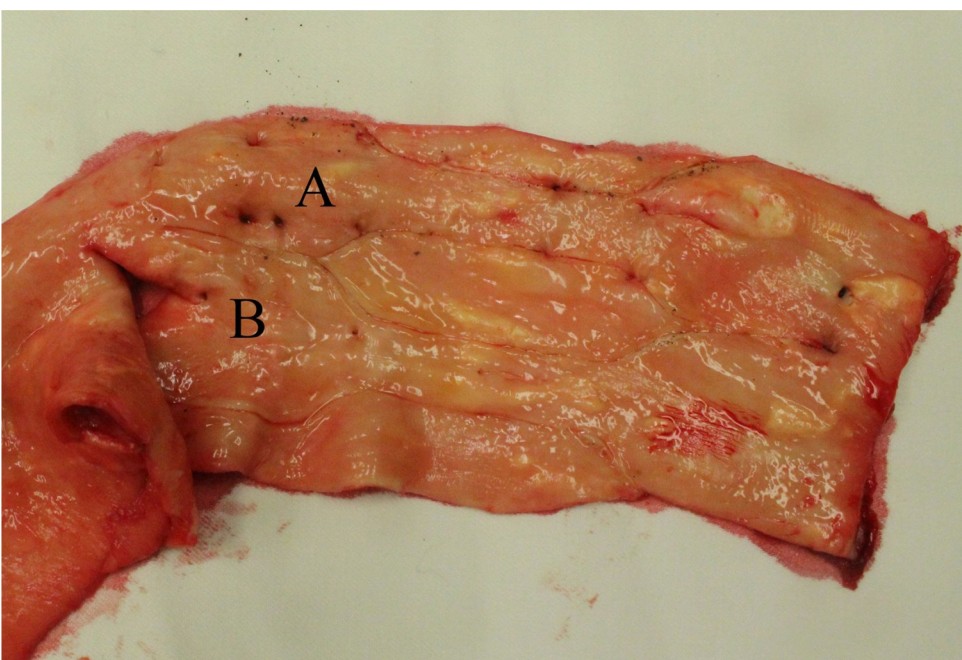

**Fig 2. Location of removed samples.** Dog-bone-shaped contour illustrates the specimen location. Specimen location from posterior wall is marked with (A), specimen location from lateral wall is marked with (B).

displacement as AFG started to show readings (so the bottom end of the rod reached the sample surface).

## Mechanical tests

Mechanical tests were performed within 48 hours after sample removal. Prior to the tests, samples were stored at 4˚C. No freezing was applied since it may interfere with the tensile properties and mechanical strength of the tissues [39]. No storing medium was used to avoid the possible effect of differing sample condition between and after sample removal. Mechanical tensile tests were performed with a Zwick/Roell Z5.0 biaxial tester (maximal load 5.0 kN, detecting accuracy 0.1 N, ZwickRoell, Ulm, Germany). The distance between the grips was 56.5 mm. The test was performed by a 500 mm/min upwards displacement (https://youtu.be/n1DczxkIYmc). The Young's modulus ($E$ (MPa)) was calculated using a linear regression procedure where the base of the elongation was the parallel length of the sample. The achieved maximal force ($F_{max}$ (N)), the elongation at the time of $F_{max}$ ($L_{max}$ (mm)), the force at the beginning of rupture ($F_{break}$ (N)), the elongation at the time of $F_{break}$ ($L_{break}$ (mm)) were registered. The point of $F_{break}$ and $L_{break}$ was determined by the force/elongation curve.

## Tissue sample classification

Tissue samples were classified by macroscopical appearance before the biomechanical measurement in the following three categories (diverse appearance were categorized as the most severe category visible) (Fig 3):

1. no atherosclerotic lesion

2. fibrous plaques

3. calcified plaques

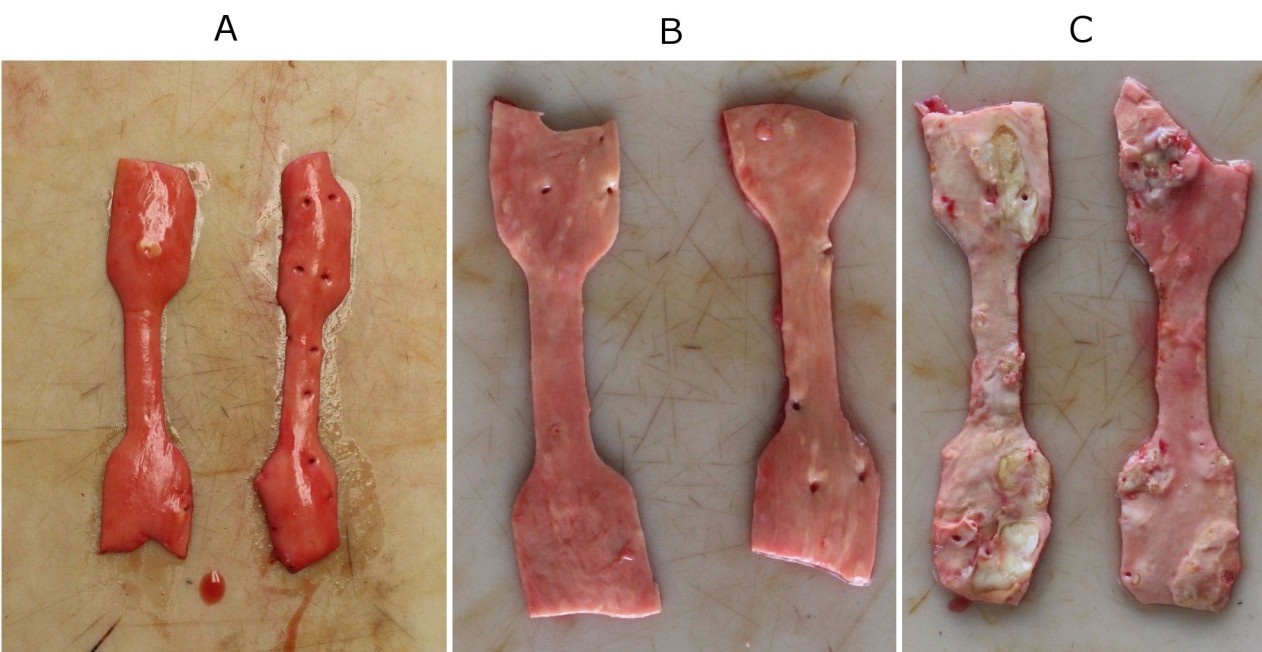

**Fig 3. Categorization by macroscopic appearance.** A: no atherosclerotic lesion (cat. 1); B: fibrous plaques (cat. 2); C: calcified plaques (cat. 3).

Tissues were fixed in 6% neutral buffered formalin, dehydrated in graded ethanol series and embedded in paraffin. Tissue sections (3 μm) were deparaffinized and rehydrated. Samples were stained with hematoxylin, bluing was performed with tap water, and eosin was added followed by dehydration with ethanol. The samples were cleared in xylene and mounted with a permanent mounting medium. Histological classification of atherosclerosis was based on the conventional classification systems proposed by American Heart Association's Committee on Vascular Lesions by Stary et al [40, 41]. Type I lesions were defined as initial or early lesions with isolated intimal foamy macrophages. Type II lesions represent multiple layers of foamy macrophages. Type III lesions are intermediate lesions with isolated extracellular lipid pools. In Type IV lesions, a lipid core could be detected, also known as atheroma. In the case of Type V lesions, the formation of prominent new fibrous connective tissue and/or calcifications could be detected (Type Va, Vb and Vc by classification of 1995 [40], and Type V., VII. and VIII by classification of 2000 [41]). Type VI lesions are complicated lesions (ulceration, hematoma or thrombosis). Samples showing diverse histological appearances (multiple lesions with different types) were categorized as the most severe lesion visible by histology.

## Statistical analysis

Statistical analyses were carried out with IBM SPSS Statistics v25 (IBM, USA).

For comparison of two related groups (posterior and lateral samples) related-samples Wilcoxon rank-sum test was used.

The comparisons of means between groups were performed with the parametric Kruskal-Wallis test. If significant difference was found between the groups, post-hoc Tukey pairwise comparison was performed with adjustment of multiple comparison.

Correlation between age and the measured parameters were calculated with Pearson correlation. Pearson correlation coefficient (r), significance and $R^2$ were calculated.

The significance level was p<0.05. The applied statistical test is always indicated in the text.

## Results

Thickness measurement was successful in all but ten samples, where the samples showed such advanced atherosclerosis, that no plaque free area was to be found to measure properly. Average thickness was 2.21 mm (SD: ±0.47, min-max: 1.6 mm—3.0 mm).

Time of death was registered with minute accuracy in all but four cases, which were not included in the statistical analysis of the association between time of death and measured parameters. The time between death and measurement, and the time between sample removal and measurement showed no correlation with registered parameters ($F_{max}$, $L_{max}$, $F_{break}$, $L_{break}$). Data is available as supplementary file (S1 File).

The comparison of the observed features of the two different locations (Group A and Group B) in the proximal descending aorta gave the following results. The measured parameters showed a significant correlation between A and B samples (Pearson's correlation: thickness p<0.001, r = 0.566,; $F_{break}$ p = 0.002, r = 0.433,; $F_{max}$ p = 0.003, r = 0.419,; $L_{break}$ p<0.001, r = 0.802; $L_{max}$ p<0.001, r = 0.795). There was no significant difference between sample groups A and B in thickness (p = 0.886, related samples Wilcoxon signed rank test) (Fig 5A), $F_{break}$ (p = 0.068) and $F_{max}$(p = 0.083) (Fig 4B and 4C). Samples from the lateral wall (B), showed significantly higher values in $L_{break}$ (p<0.001) and $L_{max}$ (p<0.001) (Fig 5D and 5E).

Macroscopically, 13.7% (14) of the cases showed no atherosclerotic lesion, fibrous plaques were observed in 24.5% (25) and calcified plaques in 61.8% (60) of samples (Table 1).

The registered curves in macroscopic groups are shown on Fig 6. There was no significant difference in thickness between groups categorized by macroscopical appearance. $F_{max}$ (p<0.001, Kruskal-Wallis test), $F_{break}$ (p<0.001). In the posterior (A) samples the difference between $L_{break}$ (p<0.001) and $L_{max}$ (p<0.001) was significant between the macroscopic group. $L_{break}$ (p = 0.009) and $L_{max}$ (p = 0.003) showed similar pattern in the lateral (B) samples (Fig 7A–7D).

Microscopically, 2% were categorized as type I lesion, 27%, 28%, 14%, 27%, respectively as Type II-V lesions, respectively (Table 2). No microscopically intact sample was found. Type VI lesions were not observed among our histological samples. Apart from different degrees of

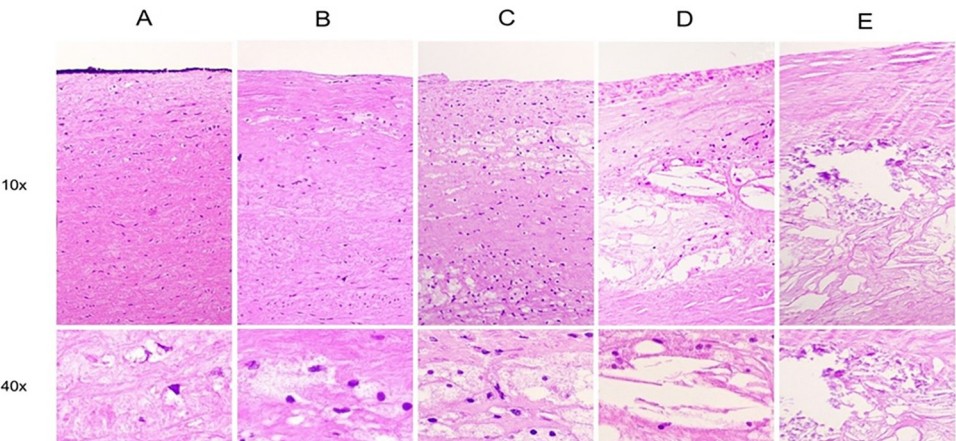

**Fig 4. Atherosclerotic lesions.** A: small, isolated groups of foamy macrophages (Type I lesion); B: foamy macrophages are stratified in adjacent layers (Type II lesion). C: multiple extracellular lipid pools (Type III lesion). D: atheroma (Type IV lesion); E: fibroatheroma–with calcification (Type V lesion). Note: type VI lesions were not observed in histological samples.

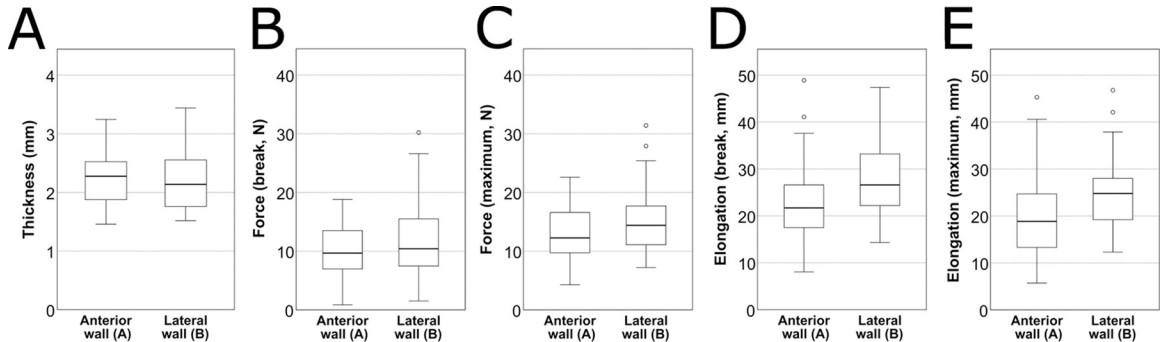

**Fig 5. Correlation of the measured parameters in sample groups A and B.** There was no significant difference between sample groups A and B in thickness (p = 0.886) (A), force at rupture ($F_{break}$) (p = 0.068) (B) and maximal force ($F_{max}$) (p = 0.083) (C). Samples from the lateral wall (B), showed significantly higher values in elongation at rupture ($L_{break}$) (p<0.001) (D) and maximal elongation ($L_{max}$) (p<0.001) (E).

atherosclerosis, no other pathological change was identified during the histological examination of the samples. The registered curves in each microscopical group are shown on Fig 8. Comparing the histological groups by the measured parameters ($F_{max}$, $L_{max}$, $F_{break}$, $L_{break}$) showed a significant difference in the means (Kruskal-Wallis test, p<0.001, p = 0.003, p<0.001 respectively) (Fig 9).

There were no significant differences in Young's modulus values between the macroscopically categorized groups (1–3) (p = 0.874, Kruskal-Wallis test) or the histological groups (1–5) (p = 0.500) (Fig 10).

By investigating the age distribution between the groups of macroscopic and histological categorization (Fig 11) we found a significant increase (Kruskal-Wallis test, p<0.001 and p<0.001).

Age showed significant negative correlation with the following four parameters (Pearson correlation) $F_{break}$ (p<0.001, $R^2$ = 0.264, r = -0.513), $F_{max}$ (p<0.001, $R^2$ = 0.292, r = -0.541), $L_{max}$ (p<0.001, $R^2$ = 0.558, r = -0.726), $L_{break}$ (p<0.001, $R^2$ = 0.558, r = -0.747) (Fig 12A–12D). No correlation was found between age and Young's modulus (p = 0.681).

## Discussion

The rupture in TAI always runs perpendicularly or near perpendicularly to the axis of the vessel [17], which indicates, that the forces affecting the longitudinally are playing the most key role in the occurrence of TAI. Our study demonstrated that atherosclerosis does affect the strength of the aortic wall in this direction. Atherosclerosis decreases the resistance of the aorta. The rupture occurs at lower force ($F_{max}$ and $F_{break}$), and at shorter elongation ($L_{max}$ and $L_{break}$) in case of the presence of atherosclerosis. This effect is most substantial if calcification is present: the resistance of aorta affected by calcification is only two-thirds on average compared to aorta affected by the early phase of atherosclerosis. This phenomenon can be clearly

**Table 1. Summarized results of groups A+B categorized by the macroscopic appearance.**

| Category (Case Number) | Mean thickness (Range, SD±) (mm) | Mean $F_{max}$ (Range, SD±) (N) | Mean $L_{max}$ (Range, SD±) (mm) | Mean $F_{break}$ (Range, SD±) (N) | Mean $L_{break}$ (Range, SD±) (mm) |
|---|---|---|---|---|---|
| 1 (14) | 2.1 (1.5–2.8, 0.3) | 16.1 (9.2–31.4, 6.1) | 28.2 (16.8–46.8, 9.6) | 13.8 (6.8–30.2, 6.6) | 31.0 (17.8–48.9, 9.4) |
| 2 (25) | 2.1 (1.5–3.4, 0.5) | 16.6 (7.9–27.9, 4.6) | 27.9 (13.4–42.1, 7.0) | 13.7 (1.58–26.6, 5.4) | 29.7 (16.3–44.8, 6.7) |
| 3 (60) | 2.4 (1.4–4.0, 0.6) | 12.3 (4.2–25.4, 3.9) | 18.4 (5.7–31.9, 5.8) | 8.8 (0.86–23, 4.5) | 21.6 (8.04–34.9, 5.6) |

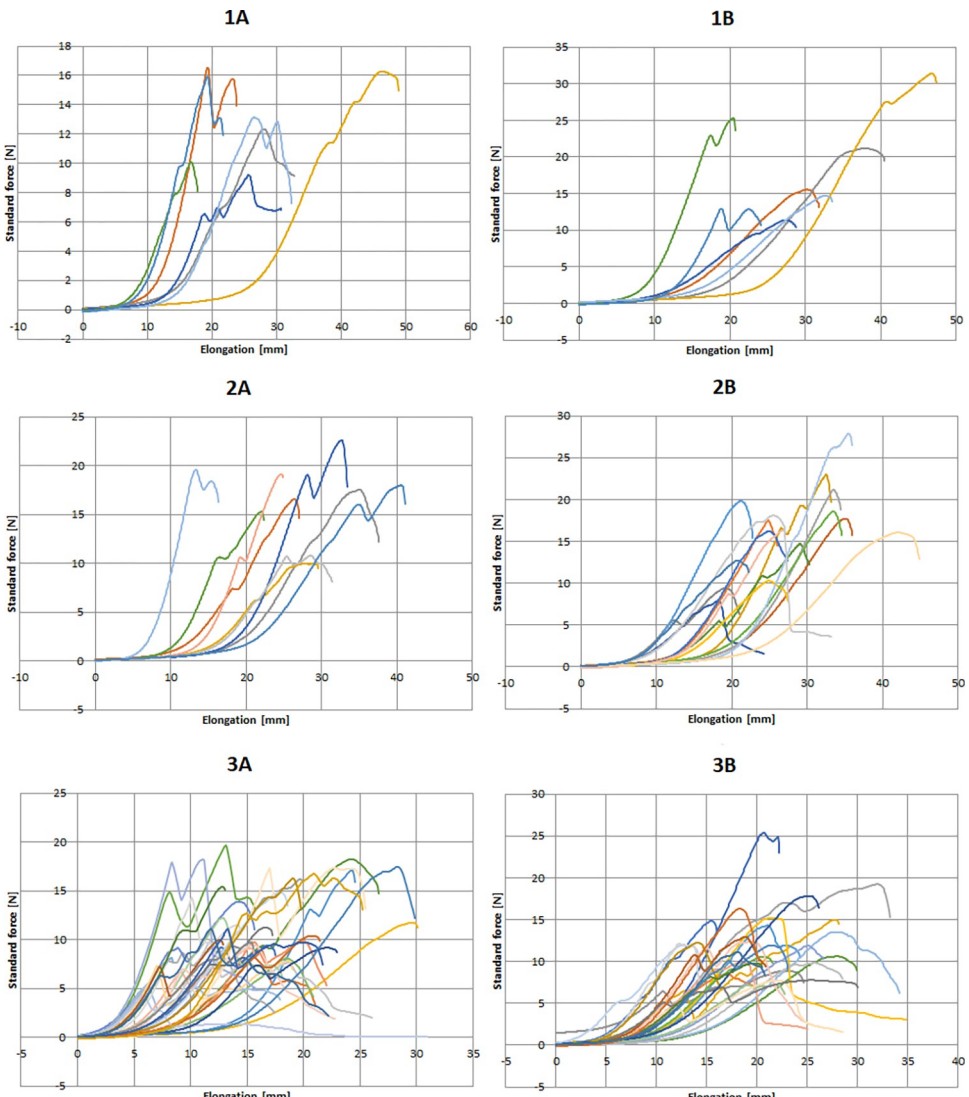

**Fig 6. The registered curves of tensile testing in different macroscopic groups.** Numbers represent the macroscopic group; letters A and B represent the sample groups (posterior and lateral wall). Force (N) is on vertical, and elongation (mm) is on horizontal axis. Each curve represents a separate specimen.

explained by the weakening structure of the tunica intima. TAI always starts from the intima [16–18], which emphasizes the role of the intima in maintaining the structural integrity of the vessel wall. It can be hypothesized, that the rupture of the intima can critically weaken the vessel wall, and therefore the rupture can spread to the other layers. However, the individual differences among cases with similar severity of atherosclerosis are still high, which can be explained by the individual differences in the above discussed microstructural factors (mostly determined by the elastic and collagen fiber content and crosslinks between fibers in tunica media [28, 29]). The data of our study correlates well with the previous study about the effect of atherosclerosis on the susceptibility of the thoracic aortic wall to delamination by Kozun et al. [31]. The association between the severity of aortic atherosclerosis and age is well established [19–21], so our results partially explain the risk of TAI increasing by age [8, 9]. However, age is an independent factor in the development of TAI. Possible explanations can be

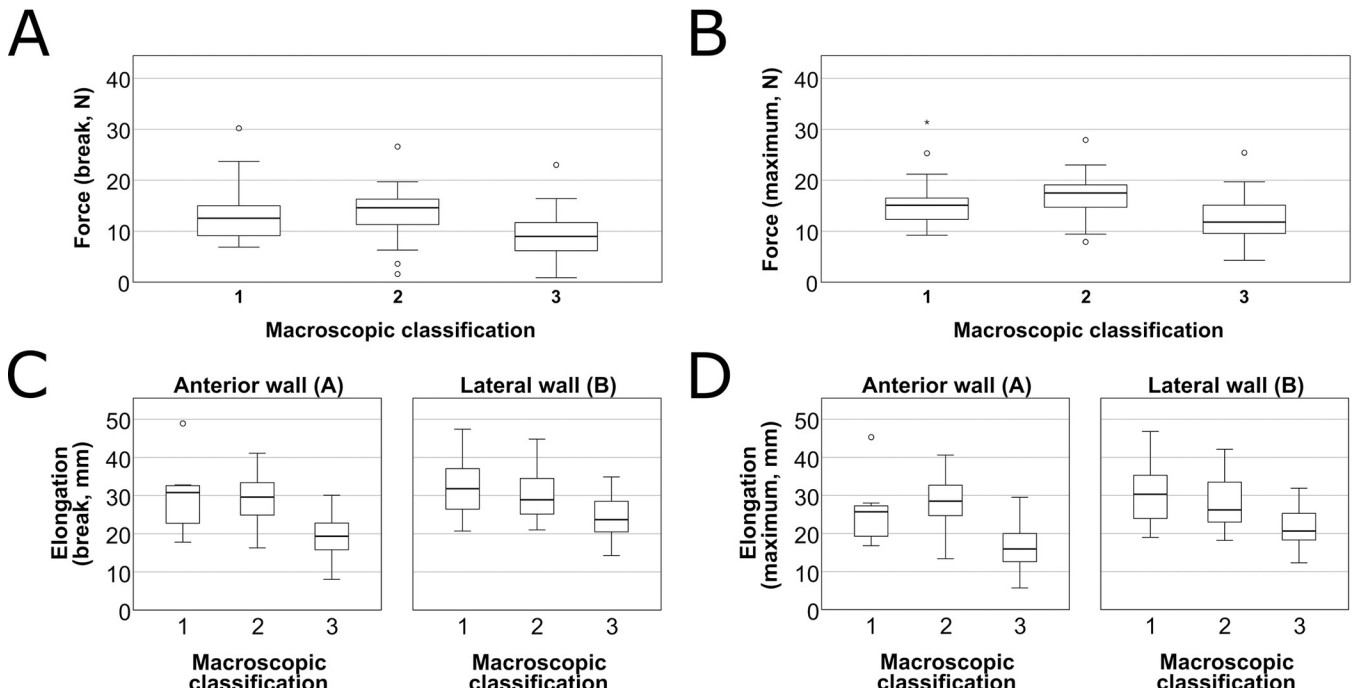

**Fig 7. The measured parameters according to the macroscopical characterization.** The horizontal axes of the graphs represent the three macroscopic groups (1: no atherosclerotic lesion, 2: fibrous plaques, 3: calcified plaques). Dots and asterisks represent outliers. **A** The force registered at the breaking point of the samples. **B** The maximum registered force. **C** Elongation of samples at breaking point by sampling site. **D** Maximum elongation of samples at breaking point by sampling site.

dilatation and other microstructural changes [26, 28, 29], but further research is necessary for a clear explanation.

An interesting finding that while $F_{max}$ and $F_{break}$ showed significant decrease with advanced atherosclerosis, the Young's modulus did not change. Young's modulus describes the elastic properties of the material, how they can withstand tension. These two results hypothesized that the calcification plaques might act as stress points or hot spots for rupture, but further studies are needed to confirm these results, because only a low number of samples belonged to the first groups with no atherosclerotic lesions. An other potential confounding factor is the relatively high age of the subjects.

It is usually not possible to examine the microstructure of the aorta in the routine assessment of traumatic aortic injuries, but macroscopical and histological assessment can be easily done. Atherosclerosis does not present uniformly in the aortic wall, so it can be advised that

**Table 2. Summarized results of groups A+B categorized by histological appearance.**

| Cat. (Number) | Mean thickness (Range, SD±) (mm) | Mean $F_{max}$ (Range, SD±) (N) | Mean $L_{max}$ (Range, SD±) (mm) | Mean $F_{break}$ (Range, SD±) (N) | Mean $L_{break}$ (Range, SD±) (mm) |
|---|---|---|---|---|---|
| **1 (2)** | 2.62 (2.3–2.9, 0.4) | 13.7 (9.9–17.5, 5.3) | 31.9 (28.9–35, 4.3) | 10.8 (9.51–12.2, 1.9) | 33.6 (29.6–37.6, 5.6) |
| **2 (27)** | 2.1 (1.5–3.4, 0.4) | 16.2 (9.1–31.4, 5.2) | 27.2 (12.6–46.8, 8.0) | 13.8 (3.6–30.2, 5.8) | 29.5 (12.9–48.9, 7.8) |
| **3 (28)** | 2.2 (1.5–3.1, 0.4) | 14.2 (7.7–25.3, 4.6) | 22.2 (11.1–42.1, 6.7) | 10.8 (1.5–23.7, 5.0) | 24.7 (15.6–44.8, 7.0) |
| **4 (14)** | 2.2 (1.4–4.0, 0.7) | 12.7 (8.0–19.2, 3.8) | 20.0 (10.1–33.5, 7.3) | 9.2 (1.9–16.4, 5.0) | 23.1 (13.1–34.6, 6.6) |
| **5 (27)** | 2.7 (1.7–4.0, 0.6) | (11.9, 4.2–25.4, 4.5) | 16.9 (5.7–29.5, 5.9) | 8.2 (0.8–23, 5.2) | 20.6 (8.0–34.2, 6) |

Note: one sample is not included due to damage to the sample during histological slide preparation.

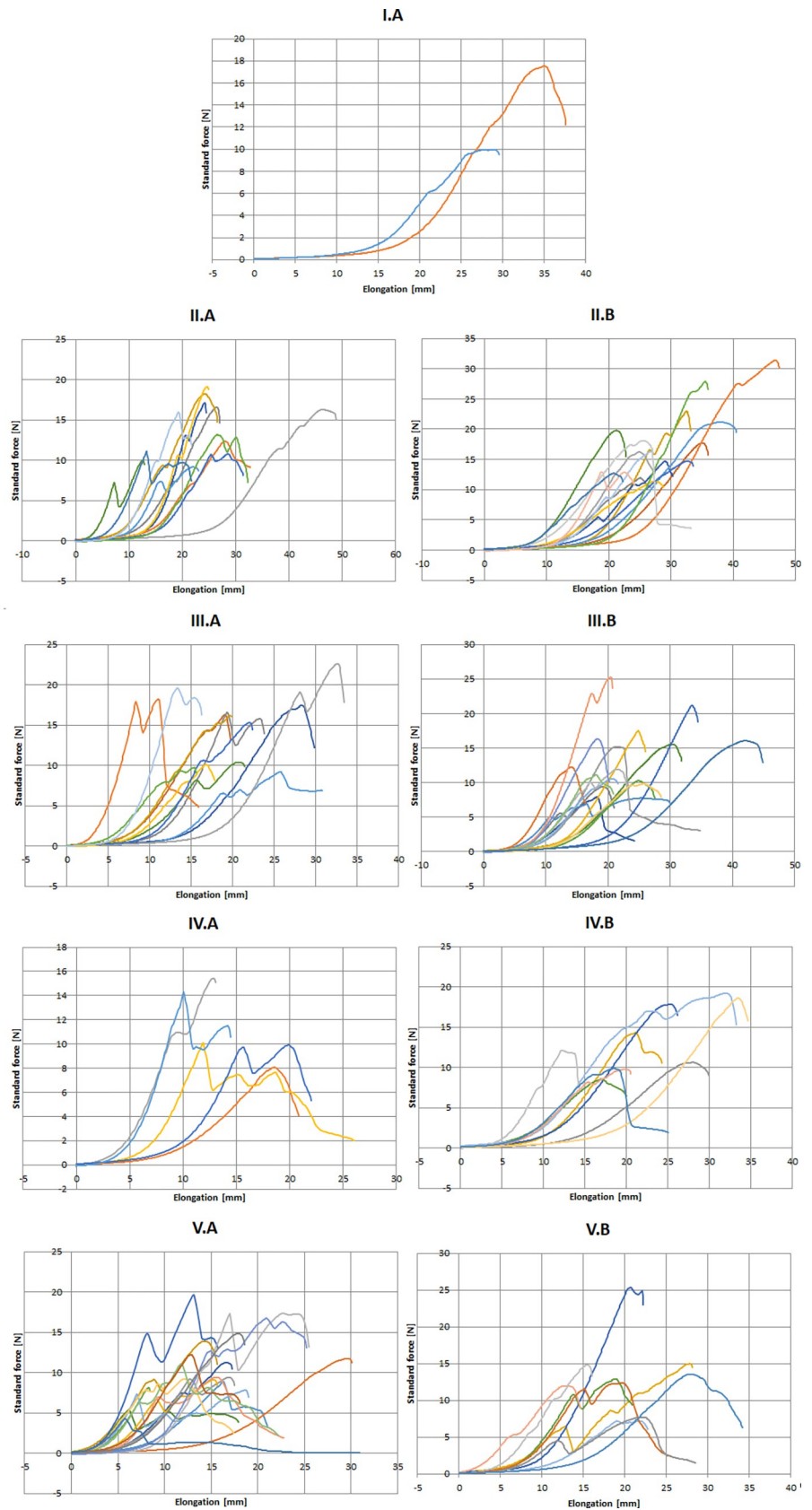

**Fig 8. The registered curves of tensile testing in different microscopic groups.** Roman numerals represent the microscopic group, letters A and B represent the sample groups (posterior and lateral wall). Force (N) is on vertical, and elongation (mm) is on horizontal axis. Each curve represents a separate specimen.

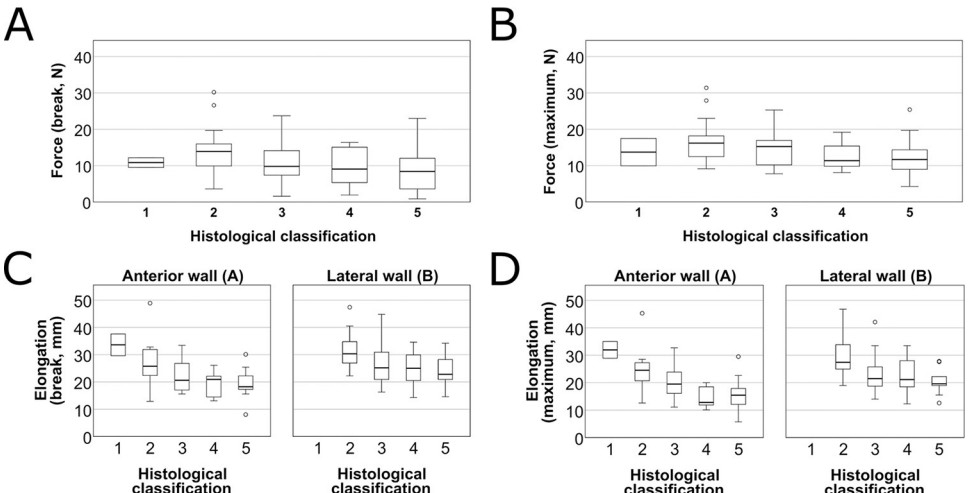

**Fig 9. The measured parameters according to the histological characterization.** The horizontal axes of the graphs represent the five histologic groups. Dots represent outliers. **A** The force registered at the breaking point of the samples. **B** The maximum registered force. **C** Elongation of samples at breaking point by sampling site. **D** Maximum elongation of samples at breaking point by sampling site.

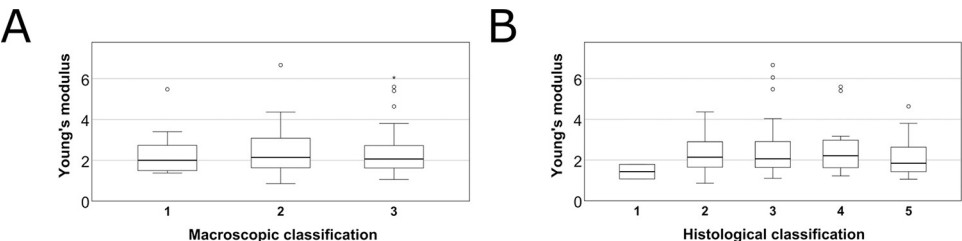

**Fig 10.** Distribution of samples Young's modulus in macroscopic (A) and histological (B) groups. Young's modulus characterizes the elasticity of the samples. It was calculated from the tensile stress and tensile strain at the linear phase of stress-strain diagram. Dots and asterixes represent outliers.

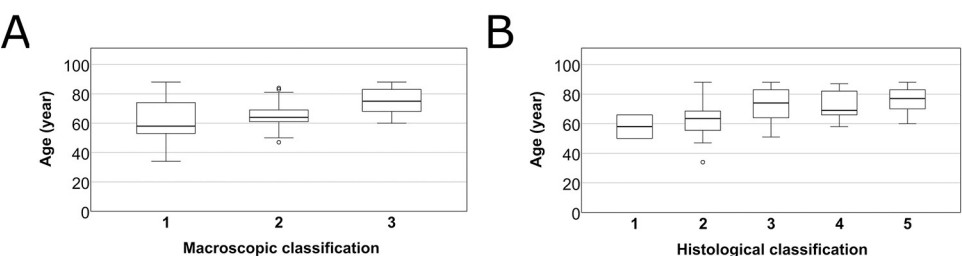

**Fig 11.** Age distribution in macroscopic (A) and microscopic (B) groups. The boxplots represent the age (years) of the subjects.

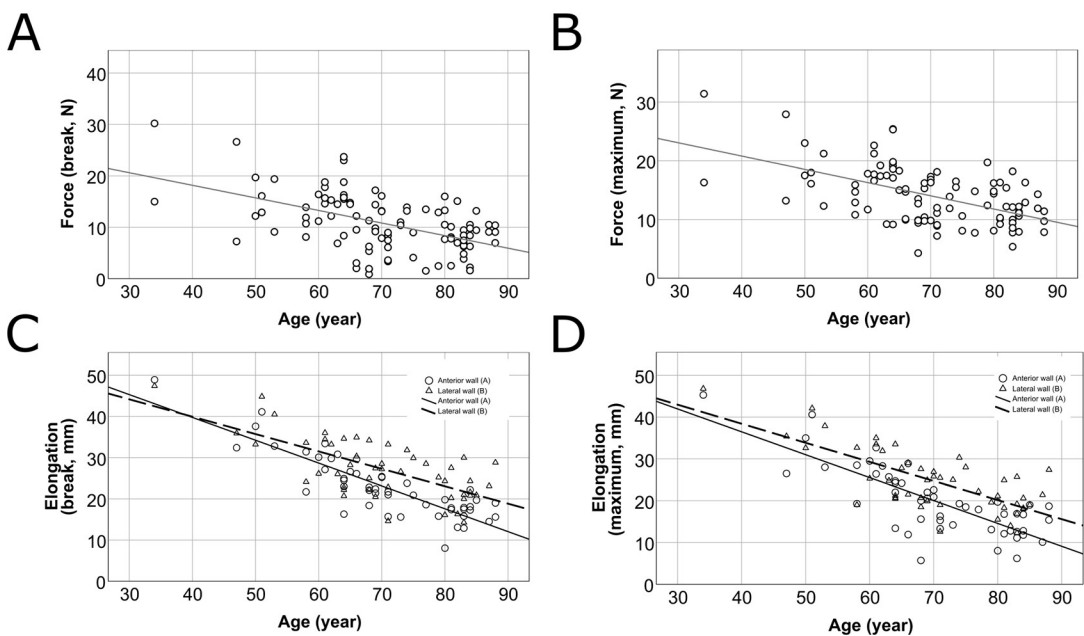

**Fig 12. Correlation between age and measured parameters.** The horizontal axes of the scatterplots show the age of the subjects. Each dot or triangle represent an individual measurement. Force at rupture ($F_{break}$) ($p<0.001$, $R^2 = 0.264$, $r = -0.513$) (A), maximal force ($F_{max}$) ($p<0.001$, $R^2 = 0.292$, $r = -0.541$) (B), maximal elongation ($L_{max}$) ($p<0.001$, $R^2 = 0.558$, $r = -0.726$) (C), elongation at rupture ($L_{break}$) ($p<0.001$, $R^2 = 0.558$, $r = -0.747$) (D).

the macroscopical and microscopical appearance of the aortic wall (severity of atherosclerosis and other vessel pathologies) at the site of the rupture should be recorded. Apart from a strict forensic pathology point of view, our observations might be translated into the clinical practice of trauma as for the call for added vigilance in case of surviving patients with high-speed deceleration mechanism injuries without signs of actual aortic injury at admission.

## Limitations

The study investigated the longitudinal resistance of the aorta in relation to atherosclerosis. Further examinations are necessary to determine this effect in the circumferential direction. The results of a uniaxial test of cross-cut aorta cannot be used for direct calculations about aortic resistance directly. Further tests (biaxial tests on circular aorta) are necessary to obtain data which can be used for more complex real-life calculations. Further investigations are required to evaluate the effect of younger age on aortic resistance to blunt trauma.

## Conclusion

Atherosclerosis affects negatively not only the biomechanical properties of the aorta but also decreases its resistance to tearing. The decrease in resistance strongly correlates with the presence of calcification.

## Supporting information

**S1 File. Statistical analysis of time-related parameters.** Electronic supplementary file containing the scatter plots, correlation coefficients and R^2 values of time between death and measurement, and the time between sample removal and measurement ($F_{max}$, $L_{max}$, $F_{break}$,

$L_{break}$). No correlation was found between the parameters.
(PDF)

**S1 Data.**
(ZIP)

## Acknowledgments

We thank János Konrád and László Mittly for their help in specimen removal.

## Author Contributions

**Conceptualization:** Roland Told, Tamás F. Molnár, Gábor Simon.

**Data curation:** Viktor Soma Poór.

**Investigation:** Dénes Pauka, Dénes Tóth, Tamás Tornóczky.

**Methodology:** Péter Maróti, Roland Told, Gábor Simon.

**Project administration:** Gábor Simon.

**Resources:** Dénes Pauka, Viktor Soma Poór.

**Supervision:** Gábor Simon.

**Visualization:** Viktor Soma Poór.

**Writing – original draft:** Gábor Simon.

**Writing – review & editing:** Viktor Soma Poór, Péter Maróti, Roland Told, Dénes Tóth, Tamás F. Molnár.

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
