## [Decision Letter · Decision Letter 0]

27 Jan 2023

PONE-D-22-25445

Role of atherosclerosis in the development of traumatic aorta injury. A biomechanical study.

PLOS ONE

Dear Dr. Simon,

Thank you for submitting your manuscript to PLOS ONE. After careful consideration, we feel that it has merit but does not fully meet PLOS ONE’s publication criteria as it currently stands. Therefore, we invite you to submit a revised version of the manuscript that addresses the points raised during the review process.

We look forward to receiving your revised manuscript.

Kind regards,

Venkateswaran Subramanian, Ph.D

Academic Editor

PLOS ONE

and https://journals.plos.org/plosone/s/file?id=ba62/PLOSOne_formatting_sample_title_authors_affiliations.pdf.

“Project no. TKP2021-NVA-06 has been implemented with the support provided from the National Research, Development and Innovation Fund of Hungary, financed under the TKP2021-NVA funding scheme. It provides support for P.M. The work is related to the Thematic Excellence Program 2021—National Excellence Sub-program; Biomedical Engineering Project (“2020-4.1.1-TKP2020”)”

Reviewers' comments:

Reviewer's Responses to Questions

**Comments to the Author**

1. Is the manuscript technically sound, and do the data support the conclusions?

Reviewer #1: No

Reviewer #2: Partly

2. Has the statistical analysis been performed appropriately and rigorously? 

Reviewer #1: No

Reviewer #2: Yes

3. Have the authors made all data underlying the findings in their manuscript fully available?

Reviewer #1: Yes

Reviewer #2: Yes

4. Is the manuscript presented in an intelligible fashion and written in standard English?

Reviewer #1: Yes

Reviewer #2: No

5. Review Comments to the Author

Reviewer #1: 1. It is not clear what the purpose of the work is. In the introduction, the authors write:

„Despite the fact that atherosclerosis has an impact on the mechanical properties of the aorta, the literature review highlighted that the connection between TAI and atherosclerosis has not been described before. The aim of our study is to investigate the possible correlation between the mechanical properties (blunt force vulnerability) of the thoracic aorta and the severity of atherosclerosis.”

Neither one nor the other assumption has been realized. The authors also did not carry out sufficient analysis to demonstrate the effect of atherosclerosis on the mechanical properties of the blood vessel wall. The authors also did not present the connection between TAI and atherosclerosis.

2. The title of the manuscript is „Role of the atherosclerosis in the development of traumatic aorta injury. A biomechanical study”. The title of the article is not adequate to the content of the work and the results. The paper presents only a general description of the mechanical properties of the aorta in the different stages of atherosclerosis. Please correct.

3. The description of the material and method is incomplete and too general.

Please include here such information as:

- age and gender of the donors

- information about the consent of the bioethics committee

- how exactly the samples were prepared? whether the vessel was cut from the body and then samples were prepared?

- an explanation of what is meant by "high-energy trauma"

- how were the samples stored? in what medium?in what temperature?

- what statistical tests were used (Statistical analysis point); why the Kruskal - Wallis test was used for statistical significance of differences between groups in each case?; please clarify. Furthermore the information about the correlation test is missing.

4. Why the authors did not reject outlier observations? What is the effect of outlier observations on the obtained values of mechanical parameters. Please clarify.

5. Are the authors sure that they determined force-strain curves? The unit of strain is not mm. Please clarify and/or correct.

6. Why did the authors cut specimens only in the longitudinal direction? Why were circumferentially cut samples not included?

7. Why did the authors adopt a VI degree scale? Currently, the VIII-grade scale is recommended.

8. On what basis did the authors use a loading speed of 500mm/min?

9. The authors determined the force-displacement curves, how was the Yonug’s modulus determined? For what range of curve? What does Emax and Ebreak mean? How did the authors calculate the values of these parameters?

10. The unit of Youngs modulus is not mm. Please correct.

11. The quality of the photos showing the graphs is too poor. Please improve.

12. The figure captions are too general. Please expand them.

13. The results point should be redo. At the beginning of this item, the authors included information about the research material and measurement methods. Please move this information to the appropriate points of the manuscript.

14. The authors stated „there were no significant differences in Young’s modulus values between the macroscopically categorized groups”. What is the reason for this? Maybe are there the calculation errors?. According to the current state of knowledge, the development of atherosclerosis affects the values of Young's modulus.

15. The introduction and discussion should be redo. At the beginnig of the introduction the authors showed state of the art on influence of atherosclerosis on mechanical properties of blood vessels. It should be presented in Introduction. Furthermore in the Discussion authors present the state of the art related to the „Structure of the aorta”, „Mechanism of blunt traumatic aortic rupture”, „Biomechanics of blunt traumatic aortic rupture”. These informations should be included in Introduction.

Reviewer #2: Summary:

I think this is an interesting paper, that presents solid work with measuring biomechanical

properties of the thoracic aorta during rupture, in specimens from forensic pathology and pathology.

The paper, however, would in my view greatly benefit from focusing both the introduction and

discussion into a clear research question and what is directly related to the paper, and also focusing the results section to with a more clear thread.

Introduction:

The introduction is very lengthy, but I think misses some key aspects into the study (for instance biomechanical aspects of atherosclerotic tissue, and what is done in tensile testing).

Lines 79 - 82: what 'receiving end' gets the proper focus. This sentence is unclear.

Methods:

line 142 - the abbreviation is not defined as customary: advanced force gauge (AFG).

line 156: this reference is about liver tissue. Is this really true for blood vessels?

Line 193: It's not clear what figure 4 refers to.

General comment: I think the manuscript would benefit from including the ethical statment in

the manuscript text (suggestion in the methods section). It would also be interesting to have

some comment on the frequency of autopsy / general indications in the authors institution.

Statistical analysis: I think the paper would benefit in clarity if the statistical analyses, were

as conventional presented in this section.

Results:

Line 211 - I think it would add to the paper to also give the thickness of the samples.

Line 214: second mention (also in methods section) that time of death was registered with a minutes accuracy -- how can this be possible?

Line 214 - 217: What are the correlations? If the authors feel that they dont belong in the main text I think they should be included as a supplementary material.

Line 218 - 224:

Here the authors need not to only give the p-values but also the actual correlations / distributions being compared.

Line 226: Fig 4, In the main text only D-E are referenced I believe, and the figure legend does not seem to correspond to the figure (these are not correlations?), and is not very descriptive.

Line 229: Calcified plaques (63) does not seem to correspond to the first column third row of Table 1.

Line 230: What registered curves are referenced here?

Table 1: I think this table would be clearer if it was transposed, and if it had also the p-values (which could then be added as a column on the right).

Line 238: This figure legend would benefit from being more descriptive to the experiment. The figure itself is nice as it shows some raw data. The resolution, however, prevents from seeing the axis labels.

Discussion:

General comments:

This discussion is very unconventionally formatted. I feel that the paper would benefit from a

much shorter discussion, that is more to the point and discusses the results of the current study, their implications in context of the literature.

6. PLOS authors have the option to publish the peer review history of their article (what does this mean?). If published, this will include your full peer review and any attached files.

Reviewer #1: No

Reviewer #2: No

---

## [Author Response · Author response to Decision Letter 0]

12 Mar 2023

Answer to Reviewers

On behalf of all the authors, I would like to thank all of you for reviewing our 

manuscript entitled “Role of the atherosclerosis in the development of traumatic aorta injury. A biomechanical study”. The remarks were valid, and greatly helped to correct the manuscript. Based on your profound reviews we think we was able to greatly increase the quality of our manuscript. We hope you will find our corrected manuscript worth publishing.

We uploaded a “track changes” version of the manuscript also (following the PlosOne guideline for revisions).

Below you can find our detailed answers to your remarks and questions.

Answers to Reviewer 1.

Reviewer #1: 1. It is not clear what the purpose of the work is. In the introduction, the authors write:

„Despite the fact that atherosclerosis has an impact on the mechanical properties of the aorta, the literature review highlighted that the connection between TAI and atherosclerosis has not been described before. The aim of our study is to investigate the possible correlation between the mechanical properties (blunt force vulnerability) of the thoracic aorta and the severity of atherosclerosis.”

Neither one nor the other assumption has been realized. The authors also did not carry out sufficient analysis to demonstrate the effect of atherosclerosis on the mechanical properties of the blood vessel wall. The authors also did not present the connection between TAI and atherosclerosis.

1.

It is important to emphasize, that the goal of this study was not to carry out a complete biomechanical study (eg. biaxial testing, examining all direction) about the aorta. The goal was to compare the vulnerability of the aorta with different level of atherosclerosis in regard of traumatic aorta injury (TAI) (the comparison has high practical value in Forensics). Therefore, the longitudinal direction was chosen as the most important in regard of TAI (see “mechanisms of TAI section”). For this purpose, we examined 104 specimens, which is far more than any other biomechanical studies did before - but the large number of specimens limited us to uniaxial testing. We added a new subsection named „study design” in the Mat&Met section for further explanation and to make our goal more clear.

The article states in the discussion section, that based on the results:

 the atherosclerosis decreases the resistance of the aorta

 the rupture occurs at lower force (Fmax and Fbreak), and at shorter elongation (Lmax and Lbreak) in case of the presence of atherosclerosis

 This effect is most substantial if calcification is present: the resistance of aorta affected by calcification is only two-thirds on average compared to aorta affected by the early phase of atherosclerosis. 

We also pointed these out in the conclusion section.

We also included in the limitations section, that further experiments are also will be necessary.

The title of the manuscript is „Role of the atherosclerosis in the development of traumatic aorta injury. A biomechanical study”. The title of the article is not adequate to the content of the work and the results. The paper presents only a general description of the mechanical properties of the aorta in the different stages of atherosclerosis. Please correct.

2.

We thought that the original title covers the manuscript content well, but - following your suggestion we changed it (as you pointed out it was maybe a little misleading). 

 The description of the material and method is incomplete and too general.

 Please include here such information as:

 - age and gender of the donors

3/a

We included the age and gender in the Mat&Met section in the original manuscript. We were only scarcely able to collect samples from younger victims (especially in females), because young victim usually die in a way we had to exclude (High energy trauma typically). This problem was also highlighted in the Limitations sections in the original manuscript.

 - information about the consent of the bioethics committee

3/b.

We did not included originally the Consent of Bioethic Comitee to the manuscript file, since PlosOne needs uploading the ethical statement separately (it can be found in the 3rd Page of the generated PDF file), and the data can give suggestion about the author’s institute, which we thought is not permitted in a blinded manuscript. However, we included it into the Mat&Met section for the revised manuscript following your suggestion.

how exactly the samples were prepared? whether the vessel was cut from the body and then samples were prepared?

3/c.

Samples was removed during the conventional autopsy. The chest complex (lung-heart-mediastinum, neck tissues and tongue) was en block removed, then the aorta was cut open longitudinally, then removed from the heart (this is the usual method of autopsy), and then the samples were removed from the aorta. We included a short reference to the autopsy method in the mat&met section for the revision.

 - an explanation of what is meant by "high-energy trauma"

3/d.

High-energy trauma can be defined for example as injuries caused by forces such as motor accidents or falls from heights, inducing extensive damage by transferring a high amount of kinetic energy to the tissue (see e.g.: 10.1016/j.ecns.2017.11.009). We included these typical examples of high energy trauma to the revised manuscript to make it more unambiguous for readers.

 - how were the samples stored? in what medium in what temperature?

3/e.

Samples were stored at 4 Celsius, no freezing was applied (this information was available at Mechanical tests section). No medium was used, since using a medium always raise the question whether it has an effect on biomechanical properties. Our research goal was to compare the vulnerability of aortas with different level of atherosclerosis on samples removed during autopsy, and using a simple cooled storing was suited better of our goals: samples were in similar conditions before the specimen removal (before the autopsy) and after the specimen removal. Using a medium or freezing would create difference in this regard.

However, statistical analyses were made to make sure, that storing condition did not had a significant effect of results. This was included in the original manuscript (results section: „The time between death and measurement, and the time between sample removal and measurement showed no correlation with registered parameters”)

We included the information about the storing medium in the revised manuscript.

what statistical tests were used (Statistical analysis point); why the Kruskal - Wallis test was used for statistical significance of differences between groups in each case?; please clarify. Furthermore the information about the correlation test is missing.

3/f.

Usually if multiple groups are compared, one would use ANOVA, but one of the presumptions of ANOVA that the data should follow a more-or-less normal distribution. Some of our subgroups are relatively small, and while the data does not seem to be skewed, formally we can not ensure normal distribution. Hence the parametric Kruskal-Wallis test was chosen. If statistically significant differences were found, all the groups were compared to each other pairwise with post-hoc Tukey comparison, where the significance values were corrected for multiple comparisons.

Why the authors did not reject outlier observations? What is the effect of outlier observations on the obtained values of mechanical parameters. Please clarify.

4. 

First, we have checked logs and notes to rule out a possible technical error. Then we checked if the values are not impossible. Because the answers were negative for both points, we decided not to remove the outliers because they might be the results of real biological variations. This is strengthened by the fact that when we get an extreme value in one sample, usually the other sample from the same subject showed similar values.

When there were outlier values, we have rerun the calculations with the outliers omitted, the results did not change significantly.

Are the authors sure that they determined force-strain curves? The unit of strain is not mm. Please clarify and/or correct.

5. 

Thank you for your remark. Yes, the strain usually is in percent. We have changed the designation to elongation.

 Why did the authors cut specimens only in the longitudinal direction? Why were circumferentially cut samples not included?

6.

Examination of circumferential direction was out of the scope of this article. Examination of the circumferential direction is essential in understanding aorta biomechanics in general, however, we wanted only to examine the effect on atherosclerosis on the resistance of aorta against traumatic aortic rupture - where the longitudinal stress is the dominant. Based on the literature, longitudinal stretching is the main mechanism of TAI, so our experiment with longitudinal direction represents the mechanism better than examine other directions.

In the future we are planning examining the circumferential direction, examining the properties in a biaxial tension, and examining the circular aorta (without cutting it up longitudinally). These tests however need different measurements (sample size, machine set up is different), so it needs to be done in a separate examination.

We want to point out, that researchers previously examining aorta biomechanics before us (eg. in multiple direction), examined much fewer samples. We examined more than hundred to have enough data for comparing different histological types.

 Why did the authors adopt a VI degree scale? Currently, the VIII-grade scale is re commended.

7.

In 1995, the American Heart Association's (AHA's) Committee on Vascular Lesions compiled much of what is known about the composition and structure of human atherosclerotic lesions, as well as the arterial sites where they develop. The statements ended with a recommendation for a numerical classification of histologically defined lesion types. Lesion morphologies that were previously classified as types Vb and Vc in the 1995 statement are now classified as lesion types VII (calcific lesion) and VIII (fibrotic lesion). No new histodiagnostical features were incorporated in the revised grading scheme. The new scale more accurately captures the potential morphological directions (evolvement) of lesion types, but that was not affecting the results of our examination. Type V in our article represents previous categorizations of Type Va-Vb-Vc and also V-VII-VIII together. It would need much more samples to examine the difference between Type V-VII and VIII.

But thank you for pointing it out, we changed that part of the article to be clearer, and also added a reference for new classification.

 On what basis did the authors use a loading speed of 500mm/min?

8.

Material testers cannot reproduce the speed (acceleration/deceleration) of high energy trauma (eg. Traffic accidents). There is also no generally accepted and used loading speed in case of biomechanical experiments on aorta. The speed we choose was determined by the machine capabilities. The Zwick/Roell Z5.0 biaxial material tester we used had a maximum of 600 mm/min.

The authors determined the force-displacement curves, how was the Yonug’s modulus determined? For what range of curve? What does Emax and Ebreak mean? How did the authors calculate the values of these parameters?

9.

Thank you for your observation. The Young's modulus was determined using the following method.

E=dσ/dε

where: dσ/dε is the slope of a least-squares regression line fit to the part of the stress/strain curve in the strain interval a < ε < b, expressed in megapascals (MPa). Since no preload was used, a and b were determined individually.

 In the article the Emax and Ebrake designations were wrong. The correct designations are Lmax and Lbreake they mean the elongation at the time of Fmax and the elongation at time of Fbreake.

 The unit of Youngs modulus is not mm. Please correct.

10.

In this case the designations were wrong, we have corrected it. See previous point.

The quality of the photos showing the graphs is too poor. Please improve.

11.

All the photos and graphs had a high quality (Checked with PlosOne PACE). They looked poor only because of the creation of PDF by the submission software. We uploaded all the figures to the following link, where you can check them how these will look like in the article:

https://ibb.co/WPnf9Qx

https://ibb.co/KNCWPFw

https://ibb.co/h9W9v1t

https://ibb.co/MVhDsK4

https://ibb.co/T2HP2HH

https://ibb.co/vQt9SYY

https://ibb.co/4pv5kqp

https://ibb.co/1TsRfdd

https://ibb.co/zRRfv2m

https://ibb.co/H40WQ8h

https://ibb.co/d5WCC7S

https://ibb.co/30jQ811

The figure captions are too general. Please expand them.

12.

The figure captions were expanded.

The results point should be redo. At the beginning of this item, the authors included information about the research material and measurement methods. Please move this information to the appropriate points of the manuscript.

13.

At first, we thought, this paragraphs fit into to results section better, but following your suggestion we moved it into the Materials and Methods section, and – revising our original opinion it seems to be better there.

The authors stated „there were no significant differences in Young’s modulus values between the macroscopically categorized groups”. What is the reason for this? Maybe are there the calculation errors?. According to the current state of knowledge, the development of atherosclerosis affects the values of Young's modulus.

14.

We have double checked the measurements and calculations, but found no typing or calculation errors. One potential explanation of this is that only a limited number of samples belonged to the first groups with atherosclerotic lesions. Another potential confounding factor is the relatively high age of the subjects. We have added mentions to these limitations to the manuscript. 

The introduction and discussion should be redo. At the beginnig of the introduction the authors showed state of the art on influence of atherosclerosis on mechanical properties of blood vessels. It should be presented in Introduction. Furthermore in the Discussion authors present the state of the art related to the „Structure of the aorta”, „Mechanism of blunt traumatic aortic rupture”, „Biomechanics of blunt traumatic aortic rupture”. These informations should be included in Introduction.

15.

Thank you for your suggestion, we moved these into the introduction. 

Answers to Reviewer 2.:

I think this is an interesting paper, that presents solid work with measuring biomechanical properties of the thoracic aorta during rupture, in specimens from forensic pathology and pathology.

The paper, however, would in my view greatly benefit from focusing both the introduction and discussion into a clear research question and what is directly related to the paper, and also focusing the results section to with a more clear thread.

Thank you for your supportive opinion, your remarks were truly useful, we modified our manuscript according your suggestions.

Introduction:

The introduction is very lengthy, but I think misses some key aspects into the study (for instance biomechanical aspects of atherosclerotic tissue, and what is done in tensile testing).

1.

Structure of the manuscript was greatly reworked following the suggestions of reviewer 1 and reviewer 2. Some unnecessary information was deleted (or shortened), and we added information about biomechanical aspects of atherosclerotic tissue and tensile testing (latter in the new “study design” subsection of Mat&Met).

We think a longer than usual introduction is necessary, since not all PlosOne readers are familiar with the forensic questions in case of blunt traumatic aortic injury. We feel that these readers would not understand the goal and relevance of our manuscript.

 Lines 79 - 82: what 'receiving end' gets the proper focus. This sentence is unclear.

2. 

The sentence wanted to refer to the explanation of why the literature data is not unambigous on risk factors of TAI. In a second look however, we found that the whole sentence is not relevant (does not adds value to the manuscript), so it was deleted in the revision.

Methods: line 142 - the abbreviation is not defined as customary: advanced force gauge (AFG).

3.

We added the missing abbreviation (AFG) in the revised manuscript.

line 156: this reference is about liver tissue. Is this really true for blood vessels?

4. 

We think that the originally referred article illustrate well, that caution is necessary with freezing of tissues which are to be used in biomechanical test. However, we decided to change the reference to an article which suggests the same in case of aortas (Yin et al). 

Line 193: It's not clear what figure 4 refers to.

5.

Figure 4 was referring to a figure of histological picture. The figure mistakenly left out from the original manuscript (we noticed it only after completing the submission). We added the missing figure and renumbered the other ones accordingly.

I think the manuscript would benefit from including the ethical statment in the manuscript text (suggestion in the methods section). It would also be interesting to have some comment on the frequency of autopsy / general indications in the authors institution.

6.

Ethical statement was included in the materials & methods section of the revised manuscript. 

An expanded ethical statement is now included in the mat&met section.

Forensic autopsy is obligatory in all cases of non-natural death in Hungary without any exceptions. Pathological autopsy is obligatory in all cases of natural death with the exception if the adherent wants to avoid it and the clinician and pathologist also agrees with it (so no new finding is excepted from autopsy). The two participating departments carry out 250 forensic and 2000 pathological autopsies annually.

We don’t think this information is relevant to the topic of the manuscript, so we did not included it in the revised version.

Statistical analysis: I think the paper would benefit in clarity if the statistical analyses, were as conventional presented in this section.

7.

The materials and methods section was updated accordingly.

Results: Line 211 - I think it would add to the paper to also give the thickness of the samples.

8. 

It’s a really good suggestion. Thickness was included separately for macroscopic and histological groups (Table 1 – Table 2), but we included the overall thickness data in the revised manuscript

 Line 214: second mention (also in methods section) that time of death was registered with a minutes accuracy -- how can this be possible?

9.

The victims we included died in hospital, where the time of death was recorded by hospital staff officially with a minute accuracy (except the 4 cases mentioned).

Line 214 - 217: What are the correlations? If the authors feel that they dont belong in the main text I think they should be included as a supplementary material.

10.

Added a supplementary file, what contains the scatter plots, correlation coefficients and R^2 values. 

Line 218 - 224:Here the authors need not to only give the p-values but also the actual correlations / distributions being compared.

11.

Added the Pearson correlation coefficients to the manuscript. Also included the scatter plots in the supplementary material.

Line 226: Fig 4, In the main text only D-E are referenced I believe, and the figure legend does not seem to correspond to the figure (these are not correlations?), and is not very descriptive.

12. 

One figure was missing from the submission. We corrected this mistake.

Line 229: Calcified plaques (63) does not seem to correspond to the first column third row of Table 1.

15.

The typo was in line 226. It was corrected to 60. 

Line 230: What registered curves are referenced here?

14.

It refers to the registered curves in the macroscopic groups. It was corrected.

Table 1: I think this table would be clearer if it was transposed, and if it had also the p-values (which could then be added as a column on the right).

16.

Table 1 was transposed to it’s place. We don’t see how p-values can fit into table 1. However, P-values are described right below the table, and fig 7 visualizes the results.

Line 238: This figure legend would benefit from being more descriptive to the experiment. The figure itself is nice as it shows some raw data. The resolution, however, prevents from seeing the axis labels.

18.

The figure legend was expanded. The resolution is not good because of the PDF generated by PlosOne submission system. See it on original resolution at this link (Fig 6 and 8):

https://ibb.co/vQt9SYY

https://ibb.co/1TsRfdd

This discussion is very unconventionally formatted. I feel that the paper would benefit from a much shorter discussion, that is more to the point and discusses the results of the current study, their implications in context of the literature.

19.

Discussion was reworked, we moved large part into the introduction section (but the transposed parts were also shortened where necessary).

We hope that we understood your remarks properly, and hope that you will find our modifications and answers satisfactory.

Yours sincerely:

Authors

---

## [Decision Letter · Decision Letter 1]

16 May 2023

PONE-D-22-25445R1Biomechanical study on the effect of atherosclerosis on the vulnerability of thoracic aorta, and it’s role in the development of traumatic aorta injury.PLOS ONE

Dear Dr. Simon,

Thank you for submitting your revised manuscript to PLOS ONE. After careful review by 2 independent reviewers, we feel that it has merit for publication in PLOS ONE with minor revision. Therefore, we invite you to submit a revised version of the manuscript that addresses the minor concern raised by Reviewer #2.

We look forward to receiving your revised manuscript.

Kind regards,

Venkateswaran Subramanian, Ph.D

Academic Editor

PLOS ONE

Journal Requirements:

Reviewers' comments:

Reviewer's Responses to Questions

**Comments to the Author**

1. If the authors have adequately addressed your comments raised in a previous round of review and you feel that this manuscript is now acceptable for publication, you may indicate that here to bypass the “Comments to the Author” section, enter your conflict of interest statement in the “Confidential to Editor” section, and submit your "Accept" recommendation.

Reviewer #1: All comments have been addressed

Reviewer #2: (No Response)

2. Is the manuscript technically sound, and do the data support the conclusions?

Reviewer #1: Yes

Reviewer #2: Yes

3. Has the statistical analysis been performed appropriately and rigorously? 

Reviewer #1: Yes

Reviewer #2: Yes

4. Have the authors made all data underlying the findings in their manuscript fully available?

Reviewer #1: Yes

Reviewer #2: Yes

5. Is the manuscript presented in an intelligible fashion and written in standard English?

Reviewer #1: Yes

Reviewer #2: Yes

6. Review Comments to the Author

Reviewer #1: I have no comments. The authors have reviewed the manuscript and they have included all comments. The manuscript is acceptable for publication.

Reviewer #2: The authors have through their changes made the paper much clearer, and have countered most of my points.

The resolution of several of the graphics are still prohibitively poor, at least when printed. Regarding question 16, transposing and 'transporting' a table is not the same, but this may not be in the authors interest to do.

In the discussion the authors write that "the rupture occurs at 2/3 of force if the aorta" is calcified" (not verbatim), and attribute this effect to "the weakening structure of the intima". I have always been under the impression that the intima is not so load-bearing in blood vessels. I think the authors could expand their reasoning here. Otherwise I do not have more comments.

7. PLOS authors have the option to publish the peer review history of their article (what does this mean?). If published, this will include your full peer review and any attached files.

Reviewer #1: No

Reviewer #2: No

---

## [Author Response · Author response to Decision Letter 1]

19 May 2023

Comment to Reviewer 1.

Thank you for your support of our manuscript, and thank you for the review, since your contribution as a reviewer greatly enhanced the quality of our manuscript.

Comment and answer to Reviewer 1.

Thank you for your support of our manuscript. We have found all your comments and suggestions well founded and helpful. Addressing the problems you highlighted and making the structural change you suggested was necessary for making a manuscript with a higher standard. Here you can found the answers to your further comments/suggestion.

The resolution of several of the graphics are still prohibitively poor, at least when printed. Regarding question 16, transposing and 'transporting' a table is not the same, but this may not be in the authors interest to do.

Graphics were created for online viewing (as Plos One is an online journal). Printing will lead to the shrinkage in size which may decrease the quality (but we think the main reason of the decrease in quality is due to how the submission system generate the pdf files – you can compare it with the original picture we gave you the links). The Figure quality was checked and approved by Plos One’s PACE tool. The table was relocated of course, not transposed.

In the discussion the authors write that "the rupture occurs at 2/3 of force if the aorta" is calcified" (not verbatim), and attribute this effect to "the weakening structure of the intima". I have always been under the impression that the intima is not so load-bearing in blood vessels. I think the authors could expand their reasoning here. Otherwise I do not have more comments.

The intima is not the main load-bearing part of blood vessels, but still it has an important function also in this regard. It is proven by the previous observation by different authors (citation 16, 17, 18) that the aortic rupture due to blunt force always starts in the intima. We can hypothesize, that the rupture of the intima decrease the integrity of the vessel wall at that point, where the rupture can continue (spread) easily to the other layers. We expanded the reasoning in the discussion section.

---

## [Decision Letter · Decision Letter 2]

12 Jun 2023

Biomechanical study on the effect of atherosclerosis on the vulnerability of thoracic aorta, and it’s role in the development of traumatic aorta injury.

PONE-D-22-25445R2

Dear Dr. Gábor Simon,

We’re pleased to inform you that your manuscript has been judged scientifically suitable for publication and will be formally accepted for publication once it meets all outstanding technical requirements.

Kind regards,

Venkateswaran Subramanian, Ph.D

Academic Editor

PLOS ONE

Additional Editor Comments (optional):

Reviewers' comments:

Reviewer's Responses to Questions

**Comments to the Author**

1. If the authors have adequately addressed your comments raised in a previous round of review and you feel that this manuscript is now acceptable for publication, you may indicate that here to bypass the “Comments to the Author” section, enter your conflict of interest statement in the “Confidential to Editor” section, and submit your "Accept" recommendation.

Reviewer #2: All comments have been addressed

2. Is the manuscript technically sound, and do the data support the conclusions?

Reviewer #2: Yes

3. Has the statistical analysis been performed appropriately and rigorously? 

Reviewer #2: Yes

4. Have the authors made all data underlying the findings in their manuscript fully available?

Reviewer #2: Yes

5. Is the manuscript presented in an intelligible fashion and written in standard English?

Reviewer #2: Yes

6. Review Comments to the Author

Reviewer #2: The authors have no further comments to the authors, and they have in my opinion addressed all previous comments.

7. PLOS authors have the option to publish the peer review history of their article (what does this mean?). If published, this will include your full peer review and any attached files.

Reviewer #2: No

---

## [Editor Report · Acceptance letter]

15 Jun 2023

PONE-D-22-25445R2 

Biomechanical study on the effect of atherosclerosis on the vulnerability of thoracic aorta, and it’s role in the development of traumatic aorta injury. 

Dear Dr. Simon:

I'm pleased to inform you that your manuscript has been deemed suitable for publication in PLOS ONE. Congratulations! Your manuscript is now with our production department. 

Kind regards, 

on behalf of

Dr. Venkateswaran Subramanian 

Academic Editor

PLOS ONE